# Human Liver Stem Cell Derived Extracellular Vesicles Alleviate Kidney Fibrosis by Interfering with the β-Catenin Pathway through miR29b

**DOI:** 10.3390/ijms221910780

**Published:** 2021-10-05

**Authors:** Sharad Kholia, Maria Beatriz Herrera Sanchez, Maria Chiara Deregibus, Marco Sassoè-Pognetto, Giovanni Camussi, Maria Felice Brizzi

**Affiliations:** 1Department of Medical Sciences, University of Torino, 10126 Torino, Italy; sharad.kholia@gmail.com (S.K.); giovanni.camussi@unito.it (G.C.); 2Molecular Biotechnology Centre, University of Torino, 10126 Torino, Italy; mariabeatriz.herrera@unito.it (M.B.H.S.); mariachiara.deregibus@unito.it (M.C.D.); 32i3T Società Per la Gestione Dell’incubatore di Imprese e per il Trasferimento Tecnologico Scarl, University of Torino, 10126 Torino, Italy; 4Department of Neurosciences, University of Torino, 10126 Torino, Italy; marco.sassoe@unito.it

**Keywords:** extracellular vesicles, kidney fibrosis, β-catenin, stem cells, miRNA

## Abstract

Human liver stem-cell-derived extracellular vesicles (HLSC-EVs) exhibit therapeutic properties in various pre-clinical models of kidney injury. We previously reported an overall improvement in kidney function following treatment with HLSC-EVs in a model of aristolochic acid nephropathy (AAN). Here, we provide evidence that HLSC-EVs exert anti-fibrotic effects by interfering with β-catenin signalling. A mouse model of AAN and an in vitro pro-fibrotic model were used. The β-catenin mRNA and protein expression, together with the pro-fibrotic markers α-SMA and collagen 1, were evaluated in vivo and in vitro following treatment with HLSC-EVs. Expression and functional analysis of miR29b was performed in vitro following HLSC-EV treatments through loss-of-function experiments. Results showed that expression of β-catenin was amplified both in vivo and in vitro, and β-catenin gene silencing in fibroblasts prevented AA-induced up-regulation of pro-fibrotic genes, revealing that β-catenin is an important factor in fibroblast activation. Treatment with HLSC-EVs caused increased expression of miR29b, which was significantly inhibited in the presence of α-amanitin. The suppression of the miR29b function with a selective inhibitor abolished the anti-fibrotic effects of HLSC-EVs, resulting in the up-regulation of β-catenin and pro-fibrotic α-Sma and collagen type 1 genes. Together, these data suggest a novel HLSC-EV-dependent regulatory mechanism in which β-catenin is down regulated by HLSC-EVs-induced miR29b expression.

## 1. Introduction

Chronic kidney disease (CKD) is a pathology that affects billions of individuals globally [1]. Currently, treatment for end stage CKD is limited to haemodialysis and kidney transplantation, both of which are restricted due to financial restraint and/or availability of donor kidneys [2]. These limitations are a major drawback in improving overall recovery, calling for novel therapeutic approaches to ameliorate the general health and well-being of CKD patients [3].

Several nephrotoxins, including aristolochic acid (AA), can damage the kidneys, eventually leading to CKD [4]. An imbalance between the injury and local compensatory responses lead to maladaptive tissue repair, which, in turn, contributes to the progression towards fibrosis and CKD [4].

AA are a group of toxins enriched in plants of the genus *Aristolochia* and *Asarum,* commonly found worldwide [5]. These plants are regularly used as part of traditional herbal therapy for the treatment of various ailments, including weight management [6]. Clinical investigations in the 1990s led to the discovery of AA as the root cause of Chinese herbal nephropathy (linked to the use of Chinese herbs) and Balkan endemic nephropathy (due to the consumption of AA contaminated wheat in the Balkans). Both of these nephropathies are now collectively classified as aristolochic acid nephropathy (AAN) [6]. AAN is a rapidly progressing form of CKD predominantly characterised by tubular damage and atrophy accompanied by interstitial nephritis and extensive fibrosis. In addition, 40% of cases also manifest with urothelial carcinoma due to the DNA adducting properties of AA. Although extensive research was done over the years to elucidate the underlying mechanisms of AAN, no effective therapeutic regimen is yet available [5]. 

Interstitial fibrosis, which is one of the progressive symptoms of AAN, is usually defined as the excessive production of extracellular matrix (ECM) coupled with tubular atrophy, microvascular rarefaction, and tissue hypoxia. Myofibroblasts are considered the key cells in the pathophysiology of fibrosis, with a variety of stimuli that could trigger their activation [7]. In order to develop anti-fibrotic therapies, it is important to understand the signalling pathways involved in the activation of myofibroblasts and the general process of fibrosis. Several signalling pathways were linked to the development and progression of kidney fibrosis [8], best characterised as being the TGFβ canonical pathway [8]. However, the mechanisms of fibroblast activation are not fully understood, as the inhibition of the TGFβ signalling cascade does not completely abrogate the fibrotic response, therefore suggesting the involvement of other signalling mechanisms [8].

The Wnt/β-catenin signalling cascade is an evolutionary conserved, highly complex pathway [9]. Depending on the magnitude and duration of its activation, this pathway can either promote repair/regeneration or aid in the progression of disease from acute kidney injury (AKI) to CKD [9]. Once secreted, Wnt proteins interact with the Frizzled (Fzd) family of receptors and co-receptors, known as low-density lipoprotein receptor-related protein-5 or 6 (LRP-5/6) in target cells, triggering the activation of Dishevelled (Dsh/Dvl) [10]. In the canonical pathway, the activation of Dsh/Dvl through the interaction between Wnt and Fzd receptors inhibits the ubiquitination activity of the β-catenin destruction complex, leading to the dephosphorylation of β-catenin [9]. The active non-phosphorylated form of β-catenin (active β-catenin) then translocates to the nucleus, where it binds to the transcription factor T-cell factor/lymphoid enhancer-binding-factor-1 (TCF/Lef-1) to promote the transcription of target genes [9]. 

Over the years, many studies have reported the involvement of the canonical WNT/ β-catenin pathway in fibrogenesis of various tissues, including the kidneys [9,10]. Experimental models, with overexpression of exogenous WNT ligands or the sustained accumulation of β-catenin in the nucleus, suggested that the canonical pathway is crucial for triggering the fibrogenic program in fibroblasts [11]. Moreover, prolonged activation of Wnt/β-catenin in vivo after AKI accelerated disease progression to CKD, whereas inhibition of the signalling pathway prevented AKI-CKD progression [9]. Several preclinical models have also reported the involvement of the Wnt/β-catenin pathway in the development of CKD. For instance, in a model of unilateral ureteral obstruction (UUO), characterized by progressive interstitial fibrosis, He et al. [12] observed an up-regulation of Wnt family ligands and Fzd receptors, as well as an accumulation of active β-catenin in the cytoplasm and nuclei of renal tubular epithelial cells, which correlated with the up-regulation of numerous β-catenin target genes. Moreover, the delivery of the Wnt antagonist DKK-1 gene reduced the levels of active β-catenin significantly, abrogating the up-regulation of target genes such as collagen type 1 and fibronectin [12]. In addition, the involvement of endothelial glucocorticoid receptors in the regulation of the Wnt canonical pathway through the Frzb gene has recently come to light. In particular, the podocyte glucocorticoid receptors were shown to play an essential role in glomerular homeostasis, as their loss contributed towards the activation of the Wnt pathway in diabetic nephropathy [13,14]. There is therefore sufficient evidence to support the involvement of the β-catenin pathway in the progression of kidney fibrosis. However, the involvement of this pathway in the progression of AAN remains to be elucidated.

Various drugs were shown to alleviate the effects of kidney fibrosis by inhibiting several factors and pathways that are up-regulated in the progression of the disease. For instance, linagliptin, an inhibitor of dipeptidyl peptidase-4 (DPP-4), was shown to ameliorate kidney fibrosis in Streptozocin-induced diabetic CD-1 mice through the inhibition of EndMT, mainly by restoring levels of the miRNA 29 family [15]. In a progressive renal fibrosis rat model of unilateral ureteric obstruction (UUO), Abbas et al. [16] showed that the administration of the sodium-glucose linked transporter-2 inhibitor empagliflozin not only improved kidney function, but also reduced the levels of key profibrotic pathway factors, such as TGFβ1 and WNT. This reno-protective effect was attributed to klotho, which was significantly up-regulated post-treatment with empagliflozin [16]. The involvement of the Janus kinase/signal transducers and activators of the transcription (JAK/STAT) pathway in the progression of fibrosis is very well reported [17]. STAT3, a key factor in the pathway, was a target of inhibition in various models of kidney fibrosis. For instance, inhibitors such as S31-201 and Stattic were tested in kidney disease models of UUO, focal segmental glomerulosclerosis, lupus nephritis, and Alport syndrome, exhibiting anti-inflammatory and anti-fibrotic effects with improved kidney function [18]. However, more studies are required to confirm these results and assess the systemic toxicity of these inhibitors [18]. Other alternative strategies in combating kidney fibrosis include the inhibition of aberrant glycolysis, which suppressed the induction of fibronectin and collagen type I in a mouse model of UUO [19]. In addition, Srivastava et al. [20] showed, through their mouse model of diabetes, that the inhibition of aberrant glycolysis restored sirtuin3 (SIRT3) levels and suppressed the fibrogenic program in these mice. Furthermore, other drugs, such as ACE inhibitors in combination with N-acetyl-seryl-aspartyl-lysyl-proline and ARBs, were shown to act as suppressors of fibrosis [21].

miRNAs are single stranded non coding RNA molecules mainly involved in regulating gene expression [22]. Notably, miRNAs are increasingly demonstrated to act as key regulators of genes involved in the pathophysiology of various diseases, including kidney fibrosis [23,24]. One of the most widely reported miRNA is the miR29 family, which is composed of three extremely similar orthologues (miR29a, miR29b, and miR29c) that share identical seed binding sequences and the ability to bind to similar target genes [24]. There is evidence from several pre-clinical and clinical studies advocating their anti-fibrotic effects in multiple disease models [25,26,27,28].

Over the past decade, extracellular vesicles (EVs) have emerged as a new paracrine mechanism of intercellular communication. Stem cell-derived EVs have generated huge interest over the years since they exhibit stem cell-like properties without the danger of forming malignancies. Various studies have reported the regenerative properties of stem cell-derived EVs in various models of human pathology, including kidney disease [29]. For instance, mesenchymal stem cell-derived EVs (MSC-EVs) improved the recovery of damaged tubular cells by promoting proliferation and inhibiting apoptosis in a glycerol-induced mouse model of AKI [30]. Interestingly, a similar effect was observed when injecting MSCs in the same model of AKI, therefore confirming that EVs exhibit similar effects to their cells of origin [30]. In a different study [31], EVs derived from MSCs were able to alleviate the disease in a rat model of renal ischemia reperfusion injury. In addition, it was shown that preconditioning with melatonin even further enhanced the therapeutic effect of EVs [31]. Grange et al. [32] reported in a CKD murine model of diabetic nephropathy an improvement in renal function and a reversal of fibrosis following treatment with MSC-EVs. In other models of CKD involving 5/6 nephrectomy [33] and unilateral ureter obstruction [34], the administration of MSC-EVs was effective in reducing glomerulosclerosis and fibrosis.

Human liver stem cells (HLSC), discovered in 2004 [35], also exhibit remarkable therapeutic properties, which span from the regeneration of liver parenchyma in a mouse model of acetaminophen-induced liver injury [35,36] to the improvement of renal morphology and function in models of AKI [37]. Specifically, in a glycerol-induced mouse model of AKI, HLSC-EVs were able to ameliorate kidney function and reduce hyaline cast formation and tubular necrosis. An enhancement in tubular cell proliferation was also observed [37]. Similar effects were observed in a diabetic nephropathy model of CKD [32]. Recently, we also reported antifibrotic and regenerative effects of HLSC-EVs in a mouse model of AAN following treatment with HLSC-EVs [38]. Furthermore, transcriptomic analysis of mouse kidney tissues revealed regulation of various inflammatory and pro-fibrotic pathways including the WNT/β-catenin pathway following HLSC-EV administration [38]. 

In this study, we investigated the involvement of the β-catenin pathway in AAN and elucidate the mechanism of action through which HLSC-EVs exert their anti-fibrotic and regenerative effects in an in vivo and in vitro CKD model of AAN.

## 2. Results

### 2.1. Characterisation of HLSC-EVs

MACSPlex EV analysis revealed that HLSC-EVs express typical mesenchymal surface markers, such as CD29 and CD44, as well as the exosomal markers CD81, CD63, and CD9, as described in [37] (Figure 1A). Western blot analyses of EV proteins also confirmed the presence of exosomal markers such as CD63 and Alix (Figure 1B). The absence of the endoplasmic reticulum protein calreticulin confirmed the purity of the EVs (Figure 1B). A further characterisation by electron microscopy revealed the presence of vesicles ranging 40–100 nm (Figure 1C). An NTA analysis of isolated HLSC-EVs demonstrated the highest concentration of particles with a size ranging between 50–110 nm (Figure 1D).

### 2.2. HLSC EVs Reduce Interstitial Fibrosis and Inflammation in AA Mice

Masson’s trichrome staining showed an increase in fibrosis reflected by the presence of collagen fibres in the interstitial space between tubules in mice treated with AA four weeks after injection (Figure 2B,D). Following treatment with HLSC-EVs, interstitial fibrosis was significantly reduced (Figure 2B,D). In addition, treatment with HLSC-EVs also reduced inflammatory cell infiltration reflected by a significant decrease in the number of CD45 positive cells, which was elevated in AA treated mice (Figure 2C,E). 

### 2.3. The Expression of Active β-Catenin and NGAL Are Reduced in AA Mice Treated with HLSC-EVs

We previously showed, through gene pathway analysis in AAN experimental mice, that the β-catenin signalling pathway is amongst the top pathways regulated by HLSC-EVs [38]. We therefore investigated the expression of active β-catenin, which is the primary component of the WNT-β-catenin pathway, in the kidneys of AAN mice (Figure 3). 

Immunohistochemical staining of formalin fixed, paraffin embedded kidney sections showed that mice treated with AA had significantly elevated levels of the activated form of β-catenin (β-catenin active) (Figure 3A,C) localised in the nuclei of damaged tubules (Figure 3B), together with the kidney tubular damage marker NGAL (neutrophil gelatinase-associated lipocalin) (Figure 3A,D), compared to healthy controls. Notably, the expression of both markers was significantly reduced following treatment with HLSC-EVs. In addition, the number of tubules expressing NGAL and β-catenin active were significantly reduced by HLSC-EVs (Figure 3E). Likewise, elevated levels of the *Ctnnb1* gene, which encodes for β-catenin, in AA mice was significantly downregulated following treatment with HLSC-EVs (Figure 3F).

### 2.4. Expression of β-Catenin in Renal Fibroblasts in an In Vitro AAN Model 

In order to investigate whether the up-regulation of β-catenin is a key factor in causing renal fibrosis, we investigated the expression of β-catenin in mouse kidney cortical fibroblasts (mkCF) in a transwell culture system (Figure 4A). Post-experimental analysis revealed a significant up-regulation of the β-catenin gene *Ctnnb1* as well as the pro-fibrotic gene *α-Sma* in fibroblasts exposed to mTECs pre-treated with AA compared to control (Figure 4B,C). Following treatment with HLSC-EVs, the expression of both genes was significantly reduced to near normal levels (Figure 4B,C). A similar trend was observed at the protein level, whereby the expression of both active β-catenin and α-SMA was significantly elevated in fibroblasts exposed to mTECs pre-treated with AA (Figure 4D,E). Again, HLSC-EVs significantly reduced β-catenin and α-SMA protein expression (Figure 4D,E), thus confirming that HLSC-EV treatment regulates β-catenin activity in fibroblasts in vitro.

### 2.5. Gene Silencing of β-Catenin Prevents the Molecular Activation of Fibroblasts in AAN In-Vitro Assay

In order to investigate the relevance of β-catenin in the activation of fibroblasts to myofibroblasts, we sought to silence the *Ctnnb1* gene that codes for β-catenin using short interfering RNA (siRNA). Briefly, mkCF cells were co-incubated with the transfection complex and *Ctnnb1* siRNA for 72 h. In selected experiments, the media was replaced at day three and cells cultured for four additional days to assess the efficiency of inhibition by the siRNA for the duration of the AAN in vitro assay. As shown in Figure 5A, we found a significant downregulation of the *Ctnnb1* gene by the respective siRNA after 72 h and 7 days post-transfection. In addition, a significant decrease of β-catenin was observed at the protein level 7 days post-transfection in fibroblasts transfected with *Ctnnb1* siRNA compared to fibroblasts transfected with control siRNA (Figure 5B). 

In a subsequent experiment, both control mkCF and mkCF silenced for β-catenin were co-incubated with mTECs pre-exposed to AA in a transwell system in the presence or absence of HLSC-EVs for 5 days at 37 °C. The results reported in Figure 6A,B revealed a significant rise in the expression of the pro-fibrotic genes *α-Sma* and *Col1a1* in the control mkCF, which was prevented following treatment with HLSC-EVs as observed previously. 

Interestingly, no significant up-regulation of *α-Sma* and *Col1a1* was observed in mkCF silenced for β-catenin following co-incubation with mTECs pre-exposed to AA. In addition, treatment with HLSC-EVs had no significant effect (Figure 6A,B). These observations confirm the importance of β-catenin in fibroblast activation in an in vitro model of AAN. 

### 2.6. HLSC-EVs Regulate β-Catenin through miR29b

The involvement of TGFβ1 in AA-induced fibrosis is well known [39], which in turn was shown to downregulate various miRNAs, including the miR29 family members [40,41]. In our in vitro model of fibrosis, whereby mkCF cells were treated with TGFβ1, a significant downregulation of all three miR29 family members was observed (Figure 7A), out of which miR29b showed a further significant downregulation compared to miR29a and c (Figure 7A). In addition, the expression of miR29b was not altered in mcKF cells following the silencing of the β-catenin gene (Figure 7B). As miR29b was shown to be involved in the regulation of pro-fibrotic genes [26,40,41], we sought to investigate the contribution of miR29b towards the antifibrotic action of HLSC-EVs. 

Real time PCR analysis of kidney tissues from AAN mice experimental groups showed a significant up-regulation of miR29b in kidney tissues of mice treated with HLSC-EVs (Figure 7C). The next step was, therefore, to investigate whether HLSC-EVs directly transferred, or induced the transcription of, miR29b in recipient cells. Hence, mkCF fibroblasts pre-seeded overnight were treated with TGFβ1 in the presence or absence of HLSC-EVs or in the presence of HLSC-EVs together with the transcription inhibitor α-amanitin. We noticed a significant downregulation of miR29b in fibroblasts following treatment with TGFβ1, which was reverted following exposure to HLSC-EVs (Figure 7D). However, treatment of HLSC-EVs, in the presence of α-amanitin, did not increase the expression of miR29b in fibroblasts (Figure 7D), suggesting that HLSC-EVs-mediated miR29b upregulation is due to induction rather than horizontal transfer.

In order to understand whether miR29b induced by HLSC-EVs plays a role in inhibiting β-catenin and pro-fibrotic genes, such as *α-Sma* and *Col1a1*, we performed the AAN in vitro assay. To this end, fibroblasts co-cultured with mTECs pre-exposed to AA were co-treated with HLSC-EVs and antimiR29b, which is known to inhibit the transcription of miR29b in cells. Co-treatment with a scramble sequence of antimiR (antimiR SCR) served as the control. As reported in Figure 7E, we found an increase in miR29b expression in fibroblasts following treatment with HLSC-EVs, which was significantly downregulated in the presence of antimiR29b. These results further confirm the contribution of HLSC-EVs in the transcription of miR29b in fibroblasts. No significant downregulation of miR29b was observed in fibroblasts co-treated with HLSC-EVs and antimiR SCR (Figure 7E). Furthermore, treatment with HLSC-EVs significantly downregulated the expression of *Ctnnb1*, *α-Sma*, and *Col1a1*, as expected. Moreover, the presence of antimiR29b nullified the effects of HLSC-EVs in downregulating *Ctnnb1* and partially inhibited the downregulation of *α-Sma* and *Col1a1* (Figure 7F). No significant inhibitory effects were observed with antimiR SCR (Figure 7F). These data identify miR29b as a potential regulator of β-catenin and a likely mechanism through which HLSC-EVs influence the expression of β-catenin and other profibrotic genes investigated in our model of AAN. 

## 3. Discussion

In the present study, we provided evidence that the Wnt/β-catenin pathway is directly involved in the process of fibrosis in a CKD model of AAN. We demonstrated that expression of the active form of β-catenin is increased in AA mice as well as in fibroblasts exposed to AA injured mTECs, and that this effect is rescued after treatment with HLSC-EVs. In addition, we reported that treatment with HLSC-EVs in vivo and in vitro also up-regulates the anti-fibrotic miRNA, miR29b. Notably, the blocking of miR29b impeded the anti-fibrotic effects of HLSC-EVs and significantly increased the expression of β-catenin and other pro-fibrotic genes. Overall, we elucidated mechanistically how HLSC-EVs regulate this pathway.

We have recently reported that both MSC-EVs and HLSC-EVs exert therapeutic effects in a CKD model of AAN [38,42]. Treatment with EVs exhibited anti-inflammatory, anti-fibrotic, and regenerative effects on the kidneys of AAN mice, improving overall kidney function. Our current data confirmed these findings, as we observed a reduction of CD45 positive cells and interstitial fibrosis in the experimental groups. Various miRNAs and genes were reported to be regulated following EV treatment, and bioinformatic analysis performed on kidney tissues revealed that the Wnt/β-catenin pathway was one of the most prominently expressed among the predicted pathways regulated by EVs [38,42]. In this study, we provided evidence for the involvement of the Wnt/β-catenin pathway in the activation of fibroblasts and the progression of fibrosis in models of AAN. 

The Wnt/β-catenin pathway is an evolutionary conserved pathway involved in regulating a variety of biological processes, including organ development and tissue homeostasis and repair. This pathway was implicated in the pathogenesis of various diseases, including cancer and cardio-vasculopathies [43]. In the mammalian kidney, Wnt/β-catenin signalling is mainly involved in nephron formation during development and becomes dormant thereafter. However, there is sufficient evidence to suggest that Wnt/β-catenin is reactivated during kidney injury [44,45]. In agreement with this, we found that exposure to AA causes increased expression of the Ctnnb1 gene and active β-catenin in damaged tubules positive for NGAL in a CKD mouse model of AAN. Furthermore, we found localisation of active β-catenin in the nuclei of AA mice. Notably, the increased expression of active β-catenin was significantly reduced by HLSC-EV treatment. This effect was further confirmed in an in vitro model of AAN, whereby activated fibroblasts treated with HLSC-EVs also expressed significantly lower levels of β-catenin and α-SMA, both at molecular and protein levels. These findings identify a novel mechanism through which HLSC-EVs regulate the progression of renal fibrosis. 

The importance of β-catenin in the activation of fibroblasts was documented in the differentiation of lung resident mesenchymal cells [46] and corneal fibroblasts [47] into myofibroblasts. A similar effect was observed in our study, in which we found a prominent up-regulation of the pro-fibrotic genes *α-Sma* and *Col1a1* in fibroblasts exposed to AA-injured mTECs. Notably, silencing of the *Ctnnb1* gene in fibroblasts did not show any significant up-regulation of the pro-fibrotic genes when co-cultured with AA-injured mTECs. Together, these findings suggest that β-catenin is crucial for the activation of myofibroblasts in different tissue types and models of injury. 

Generally abundant in healthy kidney tissue, several studies have demonstrated a reduced expression of miR29 in various animal models of fibrosis [40,48]. In addition, the disruption in the synthesis of miRNAs, such as miR29, can contribute to epithelial-to-mesenchymal transition (EMT) and endothelial-to-mesenchymal transition (EndMT) in kidney disease [49]. For instance, Srivastava el al. [49] identified a cross-talk between miR29 and miR-let-7, both of which are known regulators of TGFβ1 amongst other pro-EndMT signalling pathway, to be essential for maintaining endothelial homeostasis. A downregulation of these miRNAs, which is often observed in kidney diseases, could not only contribute towards fibrosis, but also EMT and EndMT [49]. In line with these studies, we also observed a downregulation of miR29b in mice exposed to AA. In addition, a downregulation of miR29b expression was observed in cultured fibroblasts treated with TGFβ1, in accordance with what was reported previously [50,51]. Interestingly, mice treated with HLSC-EVs expressed significantly higher levels of miR29b compared to AA mice. A similar result was also observed in vitro, whereby HLSC-EVs increased miR29b expression in fibroblasts treated with TGFβ1 or exposed to AA-treated mTECs. Moreover, this up-regulation of miR29b by HLSC-EVs was inhibited in the presence of the transcription inhibitor α-amanitin as well as the antimiR29b. Based on these data, we concluded that the up-regulation of miR29b by HLSC-EVs is primarily through induction, rather than horizontal transfer. Nonetheless, the mechanisms or EV-derived factors that regulate miR29b expression in fibroblasts remains to be clarified. Regarding pro-fibrotic genes, a significant downregulation of Ctnnb1, α-Sma, and Col1a1 was observed upon treatment with HLSC EVs in the in vitro model of AAN. Notably, the HLSC-EV effect was abrogated in the presence of antimiR29b, resulting in a significant up-regulation of Ctnnb1, α-Sma, and Col1a1. In addition, the silencing of β-catenin in vitro did not influence the expression of miR29b. Taken together, these data suggest that one mechanism by which HLSC-EVs interfere with fibrosis is linked to the induction of miR29b, which results in the inhibition of the Wnt/β-catenin pathway. 

## 4. Materials and Methods

### 4.1. Isolation and Characterisation of EVs 

EVs were isolated from the supernatant of HLSCs (2 × 10^6^ cells/T75 flask) cultured in serum-free, phenol-free Roswell Park Memorial Institute medium (RPMI) (Euroclone S.P.A, Milan, Italy) for 18 h. Viability of cells post-starvation was 98%, as confirmed by Trypan blue exclusion. The procedure for EV isolation was as follows: supernatant was first centrifuged at 3000× *g* for 15 min at 4 °C, followed by filtration through a 0.22 µm vacuum filter unit (Merk Life Sciences S.r.l, Milan, Italy) to remove cell debris and apoptotic bodies. The filtered supernatant was then ultra-centrifuged at 100,000× *g* for 2 h at 4 °C (Beckman Coulter Optima L-90 K, Fullerton, CA, USA). The resulting EV pellet obtained was resuspended in phenol-free RPMI supplemented with 1% dimethyl sulfoxide (DMSO) and stored at −80 °C until further use. Multiple batches were then pooled together and purified further by the iodixanol (Optiprep, Sigma, St. Louis, MO, USA) floating density separation protocol modified from the one described by Kowal et al. [52] to accommodate for larger centrifugation volumes. Briefly, multiple pools of EVs, acquired through ultra-centrifugation, were resuspended in 500 µL of 60% iodixanol supplemented with 0.25M sucrose. Various percentage of iodixanol working solutions (1 mL of 30%, 15%, and 5%) were layered sequentially above the EV/60% iodixanol suspension, and the final volume was adjusted to 10 mL with saline solution. The tubes were ultra-centrifuged at 350,000× *g* for 1 h at 4 °C without break in an Optima L-100K ultracentrifuge (Beckman Coulter) equipped with Type 90Ti rotor. The 15%, 30%, and 60% fractions were recovered, diluted in PBS and re-ultra-centrifuged at 100,000× *g* for 1 h at 4 °C. The pellet obtained was resuspended in PBS/1% DMSO for subsequent studies and characterisation. EVs were mainly detected in the 15% fraction, as determined by the Nanosight LM10 system (NanoSight, Wiltshire, UK), and were used for experiments. 

The characterisation of EV surface markers was performed using the human cytofluorimetric bead-based MACSPlex exosome kit (Miltenyi Biotec S.r.l, Bologna, Italy), according to manufacturer’s protocol. Briefly, three independent HLSC-EV preparations, consisting of 1 × 10^9^ EVs/preparation, were diluted in MACSPlex buffer (MPB) to a final volume of 120 µL in a 1.5 mL microcentrifuge tube. A total of 15 µL of MACSPlex exosome capture beads (containing a cocktail of 39 different exosomal marker epitopes) were then added to the tubes and incubated overnight at 4 °C on an orbital shaker at 450 rpm. The following day, the EVs were counterstained by adding 5 µL of APC-conjugated anti-CD9, anti-CD63, and anti-CD81 detection antibodies to each of the tubes and incubated for 1 hr at room temperature in the dark on an orbital shaker at 450 rpm. Post-incubation, the beads were washed once with 1 mL of MPB at 3000× *g* for 5 min, followed by a longer washing step by incubating the beads on an orbital shaker (as before) for 15 min. Post-incubation, the beads were centrifuged at 3000× *g* for 5 min and the supernatant was carefully aspirated, leaving a residual volume of 150 µL per tube for acquisition. Flow cytometric analysis was performed using the Cytoflex flow cytometer (Beckman Coulter, Brea CA, USA), whereby approximately 5000 single bead events were recorded per sample. The median fluorescence intensity (MFI) for all 39 exosomal markers were corrected for background and gated based on their respective fluorescence intensity, as per manufacturer’s instructions. 

The quantification and size distribution of purified EVs was determined by a Nanosight (NanoSight, Wiltshire, UK) equipped with a 405 nm laser. Briefly, EV preparations were diluted (1:200) in sterile saline solution and analysed by the nanoparticle analysis system using the NTA version 1.4 analytical software, as described previously [37].

### 4.2. Human Liver Stem Cells (HLSCs)

HLSCs were isolated from cryo-preserved human normal adult hepatocytes (Lonza, Basel, Switzerland) as described previously [37]. Briefly, hepatocytes were cultured in Hepatozyme-SFM medium (Lonza, Basel, Switzerland) for 2 weeks and the surviving population of cells were further cultured in α-MEM/EBM-1 (3:1) (Lonza, Basel, Switzerland) supplemented with L-glutamine (5 mM), HEPES (12 mM, pH 7.4), penicillin (50 IU/mL), streptomycin (50 μg/mL) (all from Sigma, St. Louis, MO, USA), and 10% foetal calf serum (FCS) (Invitrogen, Carlsbad, CA, USA) for expansion. The HLSCs were expanded, characterised, and cryo-preserved as described previously [37]. 

The HLSCs were positive for embryonic stem cell markers, such as nanog, oct4, sox2, and SSEA4, as well as mesenchymal stem cell markers as described in [37]. In addition, they were positive for human albumin, alpha-fetoprotein, and resident stem cell markers such as vimentin and nestin. The multipotency of the HLSCs was confirmed by osteogenic, endothelial, and hepatic differentiation under appropriate culture conditions as described earlier [37].

### 4.3. Isolation and Culture of Mouse Tubular Epithelial Cells (mTECs)

The mTECs were isolated from kidneys of healthy female C57 mice, as described before in our lab [30]. Briefly, the kidneys were minced finely with scissors and then passed through a 40-μm pore filter (Becton Dickinson, San Jose, CA); the glomeruli and tissue aggregates that remained on the surface of the filter were discarded, while the tubular cells were collected. After two washes with PBS (Lonza, Basel, Switzerland), the suspension of cells was plated and cultured in Dulbecco’s modified essential medium (DMEM) (Euroclone S.P.A, Milan, Italy) supplemented with L-glutamine (5 mM), penicillin (50 IU/mL), streptomycin (50 μg/mL), and 10% FCS. The media was changed after 5 days to eliminate dead cells [30,37]. The mTECs were characterised for their positive staining of cytokeratin, actin, alkaline phosphatase, aminopeptidase A, and megalin, and for their negative staining for von Willebrand factor, CD45, nephrin, and desmin as described previously [30,37].

### 4.4. Mouse Kidney Cortical Fibroblasts (mkCF)

Fibroblasts (mkCFs) were isolated from the kidneys of healthy male CD1 mice using a modified protocol as described previously [38]. Briefly, cortical sections of kidneys from the CD1 mice were minced and plated on gelatin-coated Petri dishes and incubated at 37 °C for 72 h in DMEM high glucose supplemented with L-glutamine (5 mM), penicillin (50 IU/mL), streptomycin (50 μg/mL), 10 mL HEPES, and 20% FCS. The medium was replaced after 72 h, and the cultures were allowed to grow for 10–14 days until a 75% confluent monolayer of fibroblast-like cells was formed. The cells were further expanded and characterised for fibroblast lineage. The media was changed twice weekly to maintain cultures. 

Fibroblasts were characterised through a series of inclusion/exclusion criteria according to their distinctive biochemical and morphological characteristics as reported previously [53]. Cells were positive for established mesenchymal markers such as: vimentin and α-SMA, as well as fibroblast specific protein 1 (FSP1), a marker of fibroblasts [38,54]. In addition, expression of the endothelial/epithelial cell marker pan-cytokeratin was found to be negative, together with minimal expression of desmin (smooth muscle cell marker), therefore confirming no contamination from these cells [38,53].

### 4.5. AAN In Vivo Model

The animal studies were conducted in accordance with the National Institute of Health Guidelines for the Care and Use of Laboratory Animals. All procedures were approved by the Ethics Committee of the University of Turin and the Italian Health Ministry (authorization number: 766/2016-PR). AAN was induced by injecting male NOD/SCID/IL2Rγ KO (NSG) mice (bred at the animal facility in the Molecular Biotechnology Centre) (6/8 weeks old; n = 12) intraperitoneally with 4 mg/kg of AA (Santa Cruz Biotechnology, Santa Cruz CA, USA) per week for 4 weeks. The HLSC-EVs (n = 9, 1 × 10^10^ EVs/mL/mouse) or vehicle alone (PBS, n = 5; as control) were administered through the tail vein 3 days after AA injections every week (Figure 2A). After 4 weeks of treatments, the mice were sacrificed and subjected to multi-parameter analyses. Immunodeficient mice were applied in this study to prevent an immunogenic reaction during repeated administration of HLSC-EVs.

### 4.6. Renal Histological Analysis

Renal histology was assessed by staining 5 µm-thick kidney sections with Masson’s Trichrome stain and analysed through microscopy. Interstitial fibrosis was quantified by measuring collagenous fibrotic areas stained in 10 random cortical fields/section from images taken at a magnification of 40× using multiphase image analysis with ImageJ software version 1.53c [55].

Immunofluorescence staining was performed as follows. Histological sections (5 μm thick) were deparaffinised, hydrated, and subjected to antigen-retrieval. The slides were blocked with 3% BSA/PBS for 30 min, permeabilised in 0.2%Triton-X100/PBS for 6 min at 4 °C, and then incubated with the primary antibodies: anti-non phospho active β-catenin (1:100, 8814s, Cell Signaling Technology, Danvers, MA, USA), anti-NGAL (1:100, ab70287, Abcam, Cambridge, UK), and anti-CD45 (1:100, ab10558, Abcam, Cambridge, UK) overnight at 4 °C. After three washes with PBS, the sections were incubated with Alexa Fluor 488 fluorescence secondary antibody (1:1000, Life technology, New York, NY, USA) and Alexa Fluor Texas red (1:1000, Life technology, New York, NY, USA) for 1 h at room temperature. Following three washes, the sections were stained with DAPI, mounted with fluorescent mounting media, and analysed via microscopy. Sections labelled with only secondary antibodies served as controls

### 4.7. Electron Microscopy

Transmission electron microscopy was performed by loading the HLSC-EVs onto 200 mesh nickel formvar carbon-coated grids (Electron Microscopy Science, Hatfield, PA, USA) for 20 min, followed by fixation in a 2.5% glutaraldehyde/2% sucrose solution. After multiple washes in distilled water, the samples were negatively stained with NanoVan (Nanoprobes Inc, Yaphank, NY, USA) and examined with a Jeol JEM 1400 Flash transmission electron microscope (Jeol, Peabody, MA, USA).

### 4.8. Transfection of mkCF Cells

The mkCF cells were transfected with validated mouse β-catenin (CTNNB1) siRNA (silencer select siRNA, I.D–s63418, Ambion Inc., Austin, TX, USA), or negative control siRNA 1 (Ambion, Inc., Austin, TX, USA) at a final concentration of 20 nM in the presence of HiPerfect transfection reagent (Qiagen, Hilden, Germany), as per the manufacturer’s instructions. Post-transfection, the cells were harvested after 72 h for RNA analysis and, after 7 days, for protein and RNA analysis. All experiments were performed in triplicate.

### 4.9. AAN In Vitro Model 

In order to study the effects of HLSC-EVs on renal cortical fibroblasts, an in vitro model of AA-induced fibrosis was organised (Figure 4A). Briefly, mTECs pre-seeded at 1.5 × 10^4^ cells/well in 24-well cell culture inserts (1.0μm pore) (Thermo Fisher Scientific, Waltham, MA, USA) were treated with 100 μM of AA for 4 h. Post-incubation, the cells were washed once with PBS and co-cultured with mkCF cells (2 × 10^4^ cells pre-seeded 24 h prior to the co-culture) in the absence or presence of HLSC-EVs (50,000 EVs/cell) for 5 days at 37 °C. In selected experiments, fibroblasts co-cultured with AA-exposed mTECs were treated with HLSC-EVs, together with 200 nM of antimiR29b or antimiR SCR (Qiagen, Hilden, Germany) in the presence of HiPerfect transfection reagent for the duration of the assay. After five days of incubation, the mkCFs were analysed for the expression of fibrotic genes by qRT-PCR. Fibroblasts co-cultured with healthy mTECs served as the controls. In order to study the mechanism of action and the activated pathways, cells treated in vitro with HLSC-EVs were subjected to both RNA and protein isolation using the Norgen all in one isolation kit (Norgen Biotek, Canada), as per manufacturer’s protocol.

### 4.10. Fibroblast α-Amanitin Assay

In order to elucidate whether HLSC-EVs transferred or induced the transcription of miR29b in mkCFs following treatment, cells were treated with the transcription inhibitor α-Amanitin (A2263, Sigma, St. Louis, MO, USA). Briefly, mkCFs (2 × 10^4^ cells/well) pre-seeded in a 24-well plate were treated with 10 ng/mL of TGFβ1 (T7039, Sigma, St. Louis, MO, USA) in combination with or without HLSC-EVs (50,000 EVs/cell) in the presence or absence of 50 µg/mL of α-amanitin for 6 h at 37 °C. Post-incubation, RNA was extracted from the cells and analysed for miR29b expression. 

### 4.11. RNA Extraction and qRT-PCR 

The total RNA was isolated from cells using TRIzol^TM^ (Ambion, Thermofisher), followed by RNA extraction using the miRNeasy mini kit (Qiagen, Venlo, Netherlands) as per the manufacturer’s protocol. Briefly, mouse renal tissues were resuspended in 1 mL of TRIzol^TM^ solution (Ambion, Thermo Fisher Scientific, Waltham, MA, USA) in microcentrifuge tubes and homogenised in a Bullet blender (Next Advance Inc, NY, USA) at a speed of 8 rpm for 3 min with 3.2 mm zirconium beads. The tubes were then incubated on an orbital shaker at 4 °C for 30 min, followed by centrifugation at 12,000× *g* for 15 min at 4 °C. The supernatant was collected and subjected to RNA isolation with the miRNeasy mini kit (Qiagen, Frederick, MD, USA), according to the manufacturer protocol. For cell RNA isolation, 700 µL of TRIzol^TM^ solution was added the cells in the wells and incubated for 10 min on a shaker. The mixture was then transferred to RNAse-free microcentrifuge tubes and subjected to RNA isolation using the miRNeasy mini kit (Qiagen, Frederick, MD, USA), according to the manufacturer’s protocol. The total RNA was quantified using the NanoDrop2000 spectrophotometer (Thermo Fisher Scientific, Waltham, MA, USA) as per the manufacturer’s protocol.

In selected experiments, total RNA and protein was extracted using the all-in-one purification kit (Norgen, Biotek Corp, Canada) as per the manufacturer’s protocol. Isolated RNA was quantified using the NanoDrop2000 spectrophotometer (Thermo Fisher Scientific, Waltham, MA, USA) and either used immediately or stored at −80 °C until further use.

The cDNA was synthesised from 200 ng of total RNA using the High-Capacity cDNA reverse transcription kit (Thermo Fisher Scientific, Waltham, MA, USA), according to the manufacturer’s protocol. A qRT PCR was performed using the StepOnePlus RT-PCR machine (Thermo Fisher Scientific, Waltham, MA, USA) in 20 µL reactions with Power SYBR Green PCR Master Mix (Thermo Fisher Scientific, Waltham, MA, USA) and specific oligonucleotide primers (Table 1) (MWG-Biotech, Eurofins Scientific, Brussels, Belgium). Data were analysed using the ΔΔCt method with *Gapdh* as endogenous control.

In selected experiments, RNA from in vivo mouse tissue and in vitro fibroblasts from an AAN in vitro assay were subjected to miRNA analysis, whereby cDNA was reverse transcribed from RNA using the miSCript Reverse Transcription kit (Thermo Fisher, Waltham, MA, USA), as mentioned above. The experiments were performed in triplicate using 3 ng of cDNA per reaction as described by the manufacturer’s protocol (Qiagen). The cDNA was subjected to RT-PCR analysis to assess the expression of miR-29b-3p, with RNU6b as endogenous control.

### 4.12. Immunoblotting

Cells were lysed in RIPA lysis buffer containing phosphatase inhibitor cocktail 2 (1:100, P5726, Sigma), phosphatase inhibitor cocktail 3 (1:100, P0044, Sigma), Phenylmethanesulfonyl fluoride (PMSF, 1:100, 93482, Sigma, St Louis, MO, USA), and protease inhibitor (1:100, P8340, Sigma, St Louis, MO, USA) for 30 min at 4 °C on a rotator. The cells were then centrifuged at 12,000× *g* for 15 min at 4 °C, and the total protein concentration was determined using the Bradford method as per manufacturer protocol.

To separate proteins, 30 µg of total cell proteins or 10 µg of EV proteins were subjected to SDS polyacrylamide gel electrophoresis (SDS-PAGE) on 4–20% gradient mini PROTEAN TGX stain-free precast gels (4568094, BIO-RAD, CA, USA). Post-electrophoresis, proteins were transferred onto Nitrocellulose membranes (IB23001, Invitrogen, Thermo Fisher Scientific, Waltham, MA, USA) using the iBlot 2 gel transfer device (IB21001, Invitrogen). This was followed by the blocking of membranes with 5% bovine serum albumin (BSA) in tris-buffered saline supplemented with 0.1% Tween-20 (TBS-T) for 1 hr at room temperature on a shaker. The blots were then incubated with the primary antibodies of interest overnight at 4 °C on a shaker. The primary antibodies included anti-β-catenin (1:1000, Ab16051, Abcam, Cambridge, UK), anti-active β-catenin (1:500, 8814s, Cell Signaling Technology), and anti-GAPDH (1:3000, Ab37168, Abcam), which served as the loading control. The appropriate secondary HRP-conjugated antibodies (1:3000, Sigma, St. Louis, MO, USA) were applied for 1 h at room temperature on a shaker, followed by three washes with TBS-T for 10 min each. Protein detection was performed using chemiluminescence-based ECL substrate (170-5061, BIO-RAD, CA, USA) using the ChemiDoc Touch imaging system (BIO-RAD, CA, USA). Relative densitometry analysis was performed using the image lab version 6.0.1 (BIO-RAD, CA, USA).

### 4.13. Statistical Analyses

Data analyses were performed using GraphPad Prism version 6.0. The results were expressed as mean ± standard deviation (SD) or the standard error of the mean (SEM) where indicated. The statistical analyses were performed by employing Student’s t-test or one-way analyses of variance (ANOVA), with a multi comparison test where appropriate. A *p* < 0.05 was considered statistically significant.

## 5. Conclusions

Overall, our findings revealed the mechanism by which HLSC-EVs exert their antifibrotic action through the suppression of Wnt/β-catenin signalling through miR29b in a model of AAN. Although the implementation of EVs as a form of alternative therapy in CKD is of great medical interest, various hurdles need to be overcome before EV therapy can be transferred from the bench to the bedside. In particular, the production of EVs at a large scale in GMP conditions is complex, and their pharmacodynamics, pharmacokinetics, and potential toxicity need to be thoroughly investigated. However, the plethora of studies that have confirmed the regenerative properties of stem cell-derived EVs in various pre-clinical models of kidney disease and other diseases outweigh the drawbacks mentioned above, paving the way towards the clinical development of EVs as an alternative therapeutic approach for multiple diseases, including CKD, in the near future.

## Figures and Tables

**Figure 1 ijms-22-10780-f001:**
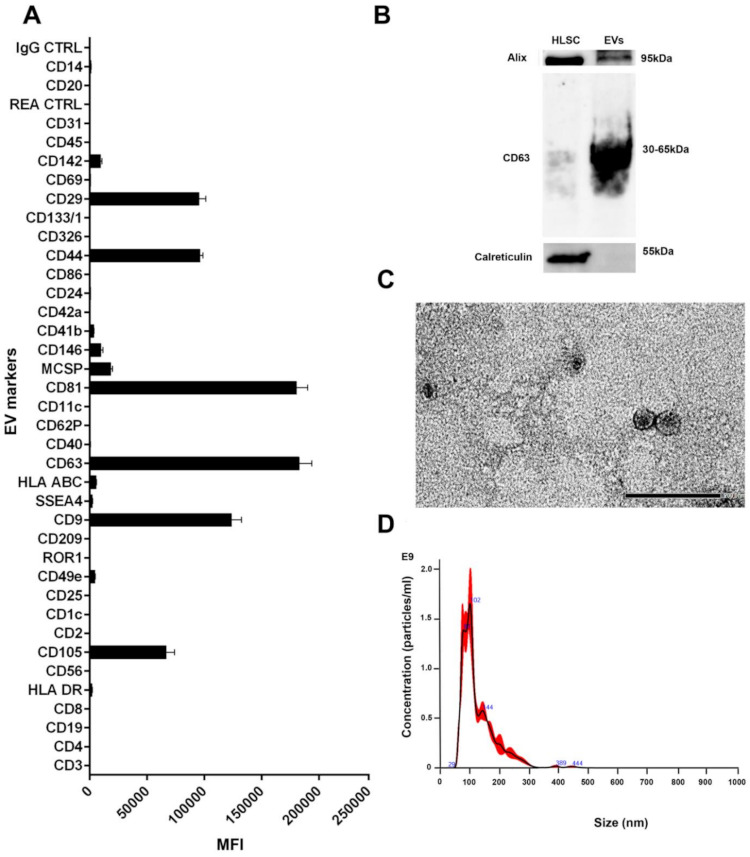
Characterisation of HLSC-EVs. (**A**) Characterisation of HLSC-EV surface antigens using a multiplex bead-based flow cytometry assay (n = 3 batches of HLSC-EVs, data represents mean ± SD). (**B**) Western blot analysis of HLSC-EVs displaying classical EV markers. (**C**) Representative electron microscopic images depicting HLSC-EV morphology (scale bar = 200 nm). (**D**) Nanoparticle tracking analysis showing the size distribution and concentration of HLSC-EVs.

**Figure 2 ijms-22-10780-f002:**
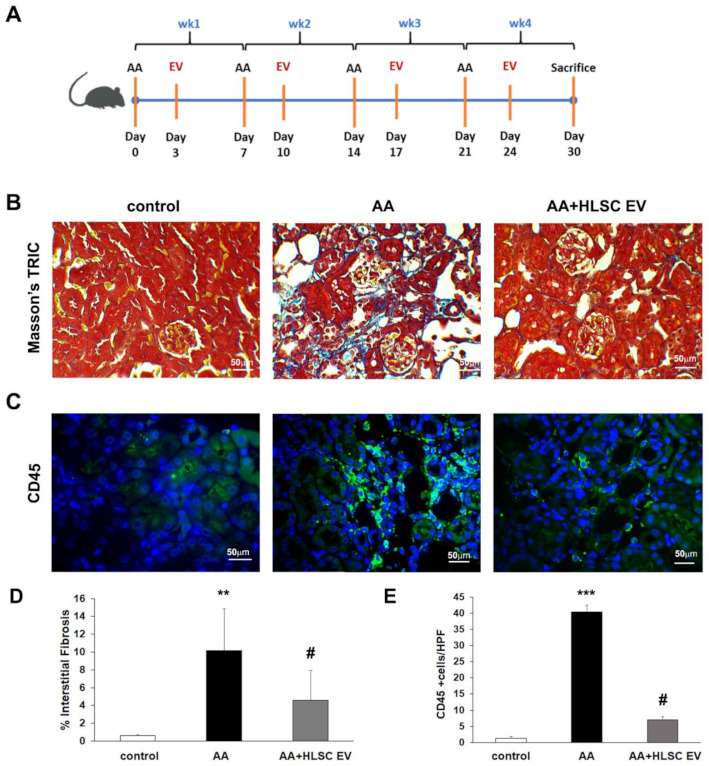
HLSC-EVs reduces fibrosis and inflammation in AAN. (**A**) Schematic representation of AAN in vivo model. (**B**) Representative micrographs of Masson’s trichome stained renal sections showing interstitial fibrosis (blue stain; scale bar = 50 µm). (**C**) Representative immunofluorescence images showing increased expression of CD45 in AAN mice and downregulation following treatment with HLSC-EVs. (**D**) Histological quantification of interstitial fibrosis by multiphase image analysis of 10 fields per section. Data represent mean ± SD; one-way ANOVA. ** *p* < 0.01 AA vs. control, # *p* < 0.01 AA+HLSC EV vs. AA. (n = 5 mice for control and AA, and n = 9 mice for AA+HLSC EV). (**E**) Histogram depicting the number of cells positive for CD45 in FFPE sections of in vivo experimental groups. Data represent mean ± SD of the number of cells positive per HPF (high power field) measure from 10 images (n = 5 mice per group). *** *p* < 0.0001 AA vs. control and # *p* < 0.01 AA+HLSC EV vs. AA.

**Figure 3 ijms-22-10780-f003:**
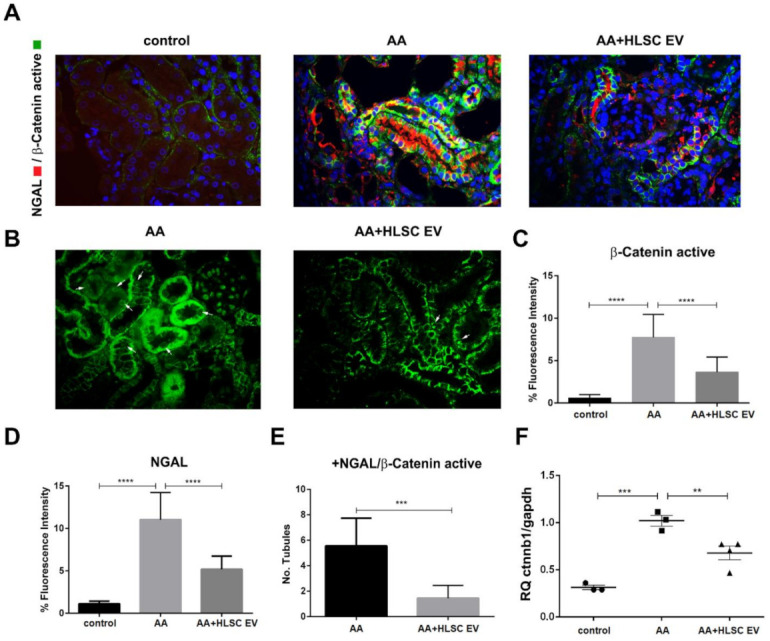
Active β-catenin and NGAL are downregulated by HLSC-EVs in AAN. (**A**) Representative immunofluorescence images show increased expression of active β-catenin and NGAL in AAN mice, which are downregulated following treatment with HLSC-EVs (AA and AA+HLSC EV) (magnification 400×). (**B**) The white arrows show nuclear localisation of β-catenin active in AA and AA+HLSC EV experimental groups (magnification 400×). (**C**) Histogram depicting the percentage fluorescence intensity of β-catenin active and NGAL (**D**) in the experimental groups. Data represent mean ± SD of the fluorescence intensity measured from 10 images per high power field taken at random (**C**,**D**) (n = 5 mice per group). **** *p* < 0.0001 AA vs. control and AA+HLSC EV vs. AA. (**E**) the number of tubules positive for colocalisation of NGAL and β-catenin active in AA and AA+HLSC EV experimental groups. Data represent mean ± SD of the number of tubules counted from 10 images per high power field taken at random (n = 5 mice per group). *** *p* < 0.001 AA+HLSC EV vs. AA. (**F**) Gene expression level of *Ctnnb1* in kidney tissue from AAN mice experimental groups. ** *p* < 0.01, *** *p* < 0.001 AA vs. control and AA vs. AA+HLSC EV. (n = 3 mice for control and AA, and n = 4 mice for AA+HLSC EV) RQ – relative quantity of *Ctnnb1* gene vs. housekeeping *Gapdh* gene.

**Figure 4 ijms-22-10780-f004:**
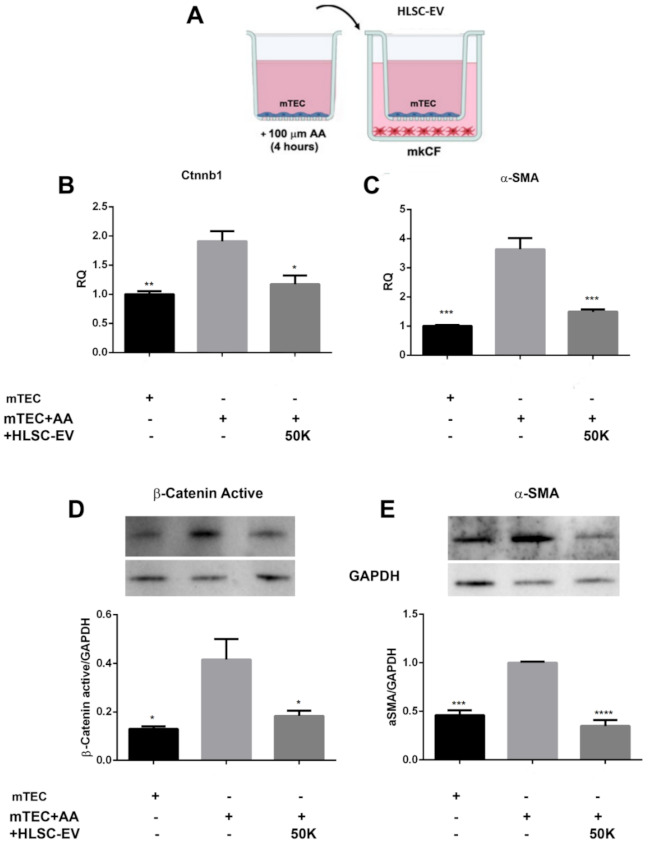
HLSC-EVs downregulate β-catenin and pro-fibrotic marker α-Sma in mkCF in an in vitro model of AAN. (**A**) A schematic representation of the transwell setting of the AAN model in vitro. Briefly, mTECs pre-treated with 100 µM AA for 4 h were co-cultured with mkCF in the presence or absence of HLSC-EVs. Post-experimental analysis revealed an up-regulation of the genes *Ctnnb1* (β-catenin) (**B**) and the profibrotic marker *α-Sma* (**C**) in mkCF exposed to AA-treated mTECs. Treatment with HLSC-EVs downregulated both genes significantly. Western blot analysis further revealed an up-regulation of active β-catenin (**D**) and α-SMA (**E**) at the protein level, which was significantly downregulated by HLSC-EVs. The data represent mean ± SEM of three independent experiments performed in quadruplicate. * *p* < 0.05, ** *p* < 0.01, *** *p* < 0.001, **** *p* < 0.0001 vs. mTEC+AA. A one-way analysis of variance with Bonferroni’s multi-comparison test was performed.

**Figure 5 ijms-22-10780-f005:**
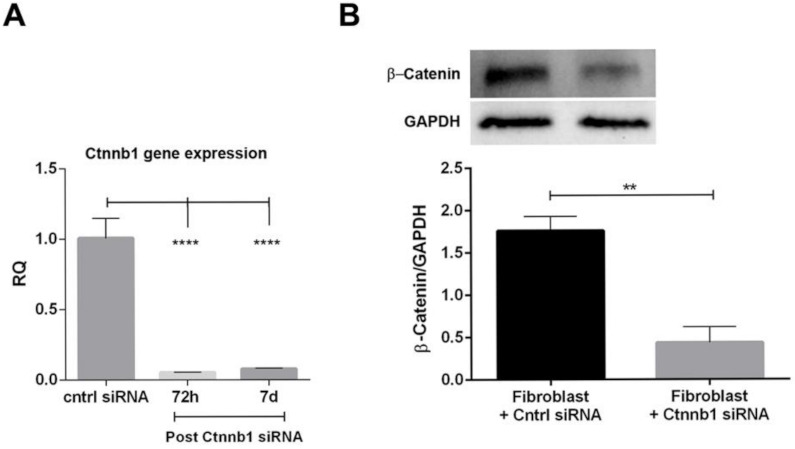
Silencing of *Ctnnb1* gene in mkCF cells. The mkCF cells were transfected with *Ctnnb1* siRNA for 72 h and then cultured for 4 further days (7 days in total post-transfection), after which the cells were subjected to molecular and protein analyses. (**A**) RT-PCR analysis revealed a significant inhibition of *Ctnnb1* gene 72 h and 7 days post-transfection compared to control siRNA. (**B**) Western blot analyses revealed a significant downregulation of β-catenin 7 days post-transfection. ** *p* < 0.01, **** *p* < 0.0001 was considered statistically significant. Data represents mean ± SEM of three independent experiments.

**Figure 6 ijms-22-10780-f006:**
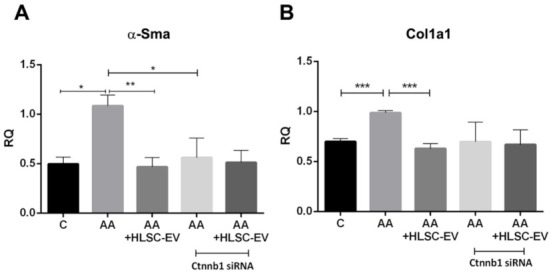
Silencing of β-catenin prevents the up-regulation of profibrotic genes in fibroblasts in vitro. mTECs pre-treated with 100 µM AA for 4 h were co-cultured with mkCF or mkCF silenced for the Ctnnb1 gene in the presence or absence of HLSC-EVs. Post-experimental analysis revealed a significant upregulation of the profibrotic genes *α-Sma* (**A**) and *Col1a1* (**B**) in fibroblasts exposed to AA-treated mTECs, which was significantly reduced in the presence of HLSC-EVs. No significant up-regulation of both profibrotic genes was observed in fibroblast silenced for the *Ctnnb1* gene, which codes for β-catenin in the presence or absence of HLSC-EVs, when compared to control. Data represent the mean ± SD of three independent experiments. A one-way ANOVA with Bonferroni’s comparison test was performed. * *p* < 0.05, *** p* < 0.01, *** *p* < 0.001 was considered to be statistically significant.

**Figure 7 ijms-22-10780-f007:**
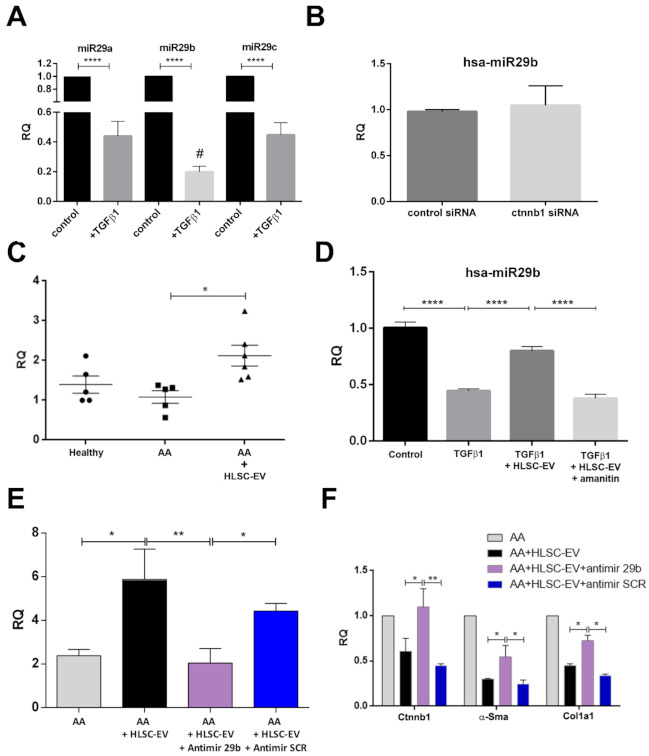
HLSC-EVs regulate β-catenin through miR29b. (**A**) Expression of hsa-miR29 a, b, and c in fibroblast treated with or without (control) TGFβ1 for 24 h at 37 °C. Data represent mean ±SD of three independent experiments performed in triplicate, **** *p* < 0.001 control vs. TGFβ1; # *p* < 0.01 TGFβ1 miR29b vs. TGFβ1 mir29a and TGFβ1 miR29c. (**B**) Expression of hsa-miR29b in mkCF cells transfected with control siRNA or *Ctnnb1* siRNA for 72 h. (**C**) Expression of miR29b in AAN mice experimental groups. A significant up-regulation of miR29b is observed in AAN mice treated with HLSC-EVs. * *p* < 0.05, (n = 5 mice per group). (**D**) RT-PCR showing the expression of miR29b in fibroblasts treated with TGFβ1 alone or TGFβ1 with HLSC-EVs, or with TGFβ1 with HLSC-EVs in the presence of α-Amanitin for 6 h at 37 °C. Post-experimental analysis revealed a significant reduction in miR29b following treatment with TGFβ1, which was up-regulated in the presence of HLSC-EVs. HLSC-EV treatment in the presence of α-Amanitin showed no significant increase in the expression of miR29b. **** *p* < 0.0001 was considered statistically significant. (**E**) The expression of miR29b in fibroblasts co-cultured with mTECs pre-treated with AA increased following treatment with HLSC-EVs and was significantly reduced in the presence of antimiR29b. * *p* < 0.05, ** *p* < 0.01 was considered to be statistically significant. (**F**) Expression of β-catenin and pro-fibrotic genes in fibroblasts co-cultured with mTECs pre-treated with AA in the presence or absence of HLSC-EVs, or HLSC-EVs together with antimiR29b, or HLSC-EVs with antimiR SCR. * *p* < 0.05, ** *p* < 0.001 were considered to be statistically significant. Data represent mean ± SD of three independent experiments.

**Table 1 ijms-22-10780-t001:** Sequences of mRNA and miRNA primers used for qRT-PCR analyses.

Gene	Primer Sequence 5’ – 3’
m_*Col1a1* forward	ATC TCC TGG TGC TGA TGG AC
m_*Col1a1* reverse	ACC TTG TTT GCC AGG TTC AC
m_*α-Sma* forward	CTG ACA GAG GCA CCA CTG AA
m_ *α-Sma* reverse	CAT CTC CAG AGT CCA GCA CA
m_*Ctnnb1* forward	GAG CAA GCT CAT CAT TCT GGC
m_*Ctnnb1* reverse	CTT CTA CAA TGG CCG GCT TG
m_*Gapdh* forward	TGT CAA GCT CAT TTC CTG GTA TGA
m_*Gapdh* reverse	TCT TAC TCC TTG GAG GCC ATG T
**miRNA**	**Primer sequence**
hsa-miR-29a-3p	TAG CAC CAT CTG AAA TCG GTT
hsa-miR-29b-3p	TAG CAC CAT TTG AAA TCA GTG TT
hsa-miR-29c-3p	TAG CAC CAT TTG AAA TCG GTT
hsa-RNU6b	CGC AAG GAT GAC ACG CAA

## Data Availability

The data that support the findings of this study are available from the corresponding author upon reasonable request.

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
