# Peer review of "Human Liver Stem Cell Derived Extracellular Vesicles Alleviate Kidney Fibrosis by Interfering with the β-Catenin Pathway through miR29b"

_ijms, 2021, doi:10.3390/ijms221910780_

Round 1
Reviewer 1 Report
The authors revealed that human liver stem cell could suppress bCatenin pathway in kidney fibrotic model. The author shown that this effects of HLSC in vivo clearly. I still have some comments to improve the manuscript. 1)Histological evaluation should be performed to assess for fibrosis and inflammation. Evaluation of fibrosis by western blotting, if possible, will provide stronger evidence. 2)AA nephropathy is a model of acute kidney injury. Please look at the degree of inflammation in the acute phase. Evaluation of macrophage infiltration, neutrophils, etc. is useful. Use KIM-1 staining as a marker of tubular damage, and look for catenin-activated tubules in the damaged tubules. 3)I didn't understand the RQ, please put it in the legend. 4)In Figure 3, it is useful to check the nuclear translocation of Catenin by extracting each protein to evaluate its activation.Author Response
We would like to thank the Reviewer for His/Her appreciation and positive comments
The authors revealed that human liver stem cell could suppress b Catenin pathway in kidney fibrotic model. The author shown that this effects of HLSC in vivo clearly. I still have some comments to improve the manuscript.
1)Histological evaluation should be performed to assess for fibrosis and inflammation. Evaluation of fibrosis by western blotting, if possible, will provide stronger evidence.
As kindly requested by the Reviewer, we provide a Masson’s staining showing interstitial fibrosis. The choice of staining allowed us to clearly identify the presence of interstitial fibrosis. We also performed a CD45 staining to demonstrate infiltration of inflammatory cells (Figure 2 of the present version of the Ms).
2)AA nephropathy is a model of acute kidney injury. Please look at the degree of inflammation in the acute phase. Evaluation of macrophage infiltration, neutrophils, etc. is useful. Use KIM-1 staining as a marker of tubular damage, and look for catenin-activated tubules in the damaged tubules.
We thank the Reviewer for the suggestion. As requested, the CD45 staining was performed (Figure 2) for evaluating leukocyte infiltration. We have also included a co-immunostaining of NGAL (as a marker of tubule damage) with active β catenin as requested by the reviewer (Figure 3A)
3)I didn't understand the RQ, please put it in the legend.
Please accept our apologies. In the present version of the Ms we have included the explanation of RQ in the legend (Figure 3)
4)In Figure 3, it is useful to check the nuclear translocation of Catenin by extracting each protein to evaluate its activation.
We thank the Reviewer for the suggestion. To better demonstrate the contribution of β-catenin signalling, its nuclear localization was evaluated in vivo (Figure 3B)
Reviewer 2 Report
The present manuscript describes extracellular vesicles from human liver stem cells and associated mechanisms in alleviating kidney fibrosis. The manuscript in general reads well and may attract some interest.
Introduction: The known drugs or effective drugs against kidney fibrosis have not been described.
- Some outstanding results in mouse models should be also included such as DPP-4 inhibitor linagliptin, empagliflozin, JAK-stat inhibitors, SIRT3, glycolysis inhibitors, and ACE inhibitors, ARBs, and peptide AcSDKP.
- Describe briefly mineralocorticoid antagonism, and the protective nature of endothelial glucocorticoid receptors, endothelial SIRT3 and endothelial FGFR1 in kidney fibrosis.
- Endothelial glucocorticoid receptor is a key molecule to regulate canonical Wnt signaling and associated fibrosis. Describe it.
- Podocyte glucocorticoid receptor regulates glomerular fibrosis through control over canonical Wnt signaling
- The authors did not describe miR-29b. miR-29b is widely studied antifibrotic microRNAs in kidneys. Antifibrotic microRNAs crosstalk between mir-29 and miR-let-7 play a critical role in regulating kidney fibrosis.
- The DPP-4 inhibitors, ACE inhibitors, and, antifibrotic peptide AcSDKP suppress kidney fibrosis by elevating the expression level of miR-29 expression.
These are the related studies missed by the authors.
Methods:
- Include complete method of TECs cells isolation from the kidney. Include purity of cell isolation
Results:
- Fig 2B: Active-beta-catenin- I did not find its nuclear localization in the IF pictures. Include both total beta-catenin and active beta-catenin staining.
- Whether nuclear beta-catenin regulates miR-29b expression or miR-29b regulates Beta-catenin levels?
- What is the effect of ctnnb silencing on miR-29 family member expression?
- What are the expression levels of other miR-29 family members like miR-29a and miR-29c? Why did the author select only miR-29b?
Discussion: miR-29b is anti-EndMT hence, include few sentences about the role of microRNAs in EMT and EndMT in kidney fibrosis.
Enhance the figure quality; figure presentation is poor.
Author Response
We thank the Reviewer for His/Her appreciation and positive comments
The present manuscript describes extracellular vesicles from human liver stem cells and associated mechanisms in alleviating kidney fibrosis. The manuscript in general reads well and may attract some interest.
Introduction: The known drugs or effective drugs against kidney fibrosis have not been described.
- Some outstanding results in mouse models should be also included such as DPP-4 inhibitor linagliptin, empagliflozin, JAK-stat inhibitors, SIRT3, glycolysis inhibitors, and ACE inhibitors, ARBs, and peptide AcSDKP.
- Describe briefly mineralocorticoid antagonism, and the protective nature of endothelial glucocorticoid receptors, endothelial SIRT3 and endothelial FGFR1 in kidney fibrosis.
- Endothelial glucocorticoid receptor is a key molecule to regulate canonical Wnt signaling and associated fibrosis. Describe it.
- Podocyte glucocorticoid receptor regulates glomerular fibrosis through control over canonical Wnt signaling
- The authors did not describe miR-29b. miR-29b is widely studied antifibrotic microRNAs in kidneys. Antifibrotic microRNAs crosstalk between mir-29 and miR-let-7 play a critical role in regulating kidney fibrosis.
- The DPP-4 inhibitors, ACE inhibitors, and, antifibrotic peptide AcSDKP suppress kidney fibrosis by elevating the expression level of miR-29 expression.
These are the related studies missed by the authors.
Please accept our apologies, all the suggested studies have been included in the introduction of the present version of the Ms
Methods:
- Include complete method of TECs cells isolation from the kidney. Include purity of cell isolation.
As kindly requested by the Reviewer, TEC isolation method has been included in the present version of the Ms (Please see Methods section).
Results:
- Fig 2B: Active-beta-catenin- I did not find its nuclear localization in the IF pictures. Include both total beta-catenin and active beta-catenin staining.
As kindly requested by the Reviewer, active β catenin staining has been included in the present version of the Ms (nuclear localization Figure 3B).
- Whether nuclear beta-catenin regulates miR-29b expression or miR-29b regulates Beta-catenin levels?
As kindly requested by the Reviewer, we checked whether β catenin can regulate miR29b. As shown in Figure 7B silencing of β catenin did not affect miR29b levels.
- What is the effect of ctnnb silencing on miR-29 family member expression?
Thank you for the question. We know that all miR29 family members are orthologues, therefore, as we did not observe any regulation of miR29b following ctnnb silencing, we speculate that ctnnb silencing should not have effect on other miR29 members (miR29a and c).
- What are the expression levels of other miR-29 family members like miR-29a and miR-29c? Why did the author select only miR-29b?
Thank you for the question. We have checked the expression level of all the miR29 family members. These data are reported in Figure 7A. As you can see all members were downregulated following treatment with TGFβ1. However, as miR29b was the most downregulated out of the three, it was selected for the study.
Discussion: miR-29b is anti-EndMT hence, include few sentences about the role of microRNAs in EMT and EndMT in kidney fibrosis.
As kindly requested by the Reviewer, this information has been included in the Discussion section and a reference included (ref- 49).
Enhance the figure quality; figure presentation is poor.
As kindly requested by the Reviewer, Figure quality has been improved.
Reviewer 3 Report
The manuscript is presenting mechanism underlying the effects of HLSC-EVs in kidney fibrosis. As the authors have shown data on anti-fibrotic effects of HLSC-EVs, some of the data seem to be missing.
1) Although the authors have reported the data on same model(AAN) in previous study in the reference (ref 22,27), to correlate that beta catenin pathway was correlated with kidney fibrosis in animal model, molecular or histologic data of in vivo model should be presented.
2) There are several downstream miRNAs of TGF-beta pathway. Also, there are several downstream pathways of beta catenin. Within MiRNA29 cluster, why the authors focused on miRNA29b?
Author Response
We thank the Reviewer for His/Her appreciation and positive comments
The manuscript is presenting mechanism underlying the effects of HLSC-EVs in kidney fibrosis. As the authors have shown data on anti-fibrotic effects of HLSC-EVs, some of the data seem to be missing.
- Although the authors have reported the data on same model(AAN) in previous study in the reference (ref 22,27), to correlate that beta catenin pathway was correlated with kidney fibrosis in animal model, molecular or histologic data of in vivo model should be presented.
As kindly requested by the Reviewer, histological data has been included in the present version of the Ms (Figure 2A).
- There are several downstream miRNAs of TGF-beta pathway. Also, there are several downstream pathways of beta catenin. Within MiRNA29 cluster, why the authors focused on miRNA29b?
Thank you for the question. We have checked the expression level of all the miR29 family members. These data are reported in Figure 7A. As you can see all members were downregulated following treatment with TGFβ1. However, as miR29b was the most downregulated out of the three, it was selected for the study.
Round 2
Reviewer 1 Report
The revisited paper has been improved enough. I do not additional comment and question.
Reviewer 2 Report
Minor english editing is required.
Thanks